# Unmanned Vessel Collision Avoidance Algorithm by Dynamic Window Approach Based on COLREGs Considering the Effects of the Wind and Wave

Xiaoyu Yuan [1] , Chengchang Tong [1], Guoxiang He [2] and Hongbo Wang [1,*]

1 State Key Laboratory on Integrated Optoelectronics, College of Electronic Science and Engineering, Jilin University, Changchun 130012, China; xyy22@mails.jlu.edu.cn (X.Y.); tongcc23@mails.jlu.edu.cn (C.T.)
2 Process Department, CSSC Marine Technology Co., Ltd., Shanghai 200136, China; hegxabc@126.com
* Correspondence: wang_hongbo@jlu.edu.cn; Tel.: +86-0431-8514-8242

**Abstract:** In recent years, the rapid development of artificial intelligence algorithms has promoted the intelligent transformation of the ship industry; unmanned surface vessels (USVs) have become a widely used representative product. The dynamic window approach (DWA) is an effective robotic collision avoidance algorithm; however, there are deficiencies in its application to the ship field. First, the DWA algorithm does not consider International Regulations for Preventing Collisions at Sea (COLREGs), which must be met for ship collision avoidance to ensure the navigational safety of the USV and other ships. Second, the DWA algorithm does not consider the influence of wind and waves on the collision avoidance of USVs in actual navigational environments. Reasonable use of windy and wavy environments not only improves navigational safety but also saves navigational time and fuel consumption, thereby improving the economy. Therefore, this paper proposes an improvement algorithm by DWA referred to as utility DWA (UDWA) based on COLREGs considering the sailing environment. The velocity sampling area was improved by dividing the priority, and the velocity function in the objective function was enhanced to convert the effect of wind and waves on the USVs into a change in velocity. The simulation results showed that the UDWA algorithm optimized the distance to the obstacle ship by 43.25%, 31.36%, and 67.81% in a head-on situation, crossing situation, and overtaking situation, respectively, compared to the COLREGs-compliant DWA algorithm, which considers the COLREGs. The improved algorithm not only follows the COLREGs but also has better flexibility in emergency collision avoidance and can safely and economically navigate and complete collision avoidance in windy and wavy environments.

**Keywords:** autonomous navigation; dynamic window approach; unmanned surface vehicle; International Regulations for Preventing Collisions at Sea (COLREGs)

## 1. Introduction

Unmanned surface vessels (USVs) are intelligent ships that can be remotely controlled or autonomously operated. With the progress and development of science and technology, USVs have recently received widespread attention in hazardous application scenarios, such as ocean exploration, rescue, and military reconnaissance [1–6]. According to the mission requirements, USVs must be able to identify their current position, control the propellers to move autonomously toward the destination, and identify and avoid static and dynamic obstacles during movement. In particular, path planning for USVs to avoid dynamic and static obstacles and navigate safely and autonomously from the starting point to the target point is needed.

Path planning can be divided into two categories: global path planning and local path planning. Global path planning belongs to static planning algorithms [7–9], and the existing map information (i.e., simultaneous localization and mapping) serves as the basis for determining the optimal path from the starting point to the target point. The

implementation of global path planning usually includes graph search algorithms, such as Dijikstra's algorithm [10], A* algorithm [11], rapidly exploring random tree (RRT) algorithm [12], and intelligent algorithms, such as ant colony optimization (ACO) [13] and genetic algorithm (GA) [14]. Meanwhile, local path planning belongs to dynamic planning algorithms, where USVs first perceive the surrounding environment based on their own sensors and then plan a route required for its safe traveling; this approach is often used in ship encounters, obstacle avoidance, and other scenarios. Usually, the implementation of local path planning includes the dynamic window algorithm (DWA) [15], artificial potential field (APF) [16], and Bessel curve algorithm. Consequently, some scholars have proposed neural networks (NN) [17] and other intelligent algorithms. In addition, there are some path planning algorithms that utilize AIS data. Ref. [18] proposes the multi-objective peak DP algorithm (MPDP), which uses a peak sampling strategy that takes into account the three optimization objectives of trajectory and adds an obstacle detection mechanism to achieve a compression algorithm that is more suitable for curved trajectories. Ref. [19] defines two indicators to evaluate the navigation collision risk: the degree of velocity obstacle intrusion (DVOI) and time of velocity obstacle intrusion (TVOI). These two indicators assess the risk of collision, respectively, from two aspects speed and course. In addition, a method for screening the collision avoidance operation points in ship AIS trajectories using a trajectory compression algorithm is proposed.

DWA is a mature and effective local path-planning algorithm used to avoid collision targets by reflecting the dynamic state of USVs. DWA can be applied fused with global path control algorithms, resulting in its wide application in mobile robots. The principle of the DWA algorithm is to determine the velocity sampling space or dynamic window (three limits), sampling multiple sets of velocities in the velocity space, simulating their trajectories in a certain time, scoring these trajectories by an evaluation function, and selecting the optimal trajectory to drive the USV motion. However, general DWA algorithms only consider collision avoidance but do not regard the effects of weather factors or the need for USVs to comply with the International Regulations for Preventing Collisions at Sea (COLREGs) [20]. Therefore, traditional DWA algorithms, which can still plan paths for ships in dangerous situations [21], are suggested. In the combination of the global path-planning RRT algorithm and the DWA algorithm considering second-order nonlinear constraints [22], the path generated by the A* algorithm is first smoothed to reduce waypoints and generate shorter paths; the resulting A* algorithm is used combined with the DWA algorithm. Ref. [23] proposed a novel fuzzy control path planning algorithm based on scale factors. The proposed algorithm combines a stability fuzzy controller with a collision risk controller to carry out adaptive control of DWA. Ref. [24] used circle chaotic mapping, adaptive weight factor, and the simplex method to improve the initial solution and spatial search efficiency and accelerate the convergence of the algorithm. Optimal path information planned by the improved WSO is put into the DWA to enhance the USV's navigation performance. An improved A-star algorithm for USV path planning and improved dynamic window approach (IDWA) for collision avoidance were proposed. The improved A-star algorithm was introduced to let the USV avoid static obstacles and reach its destination without being trapped in local optimization [25].

Although some studies include COLREGs, its consideration is not complete, resulting in ship collision even with collision avoidance. Ref. [26] used the A* algorithm as a global path-planning algorithm and the DWA algorithm as the local path-planning algorithm to avoid dynamic obstacles by tracking the local target point and following the global path until the final target point, as shown in Figure 1. This approach has the novelty of adding a sea-state factor to the objective function. As the sea level increases, the weight of the ship speed function decreases, the distance function increases, the actual sailing speed decreases, and the distance between the ship and the obstacle increases. Conversely, as the sea level decreases, the weight of the ship speed function increases, the distance function decreases, the actual sailing speed increases, and the distance between the ship and the nearest obstacle decreases. Ref. [27] complies with the COLREGs rules by disabling

different speed sampling spaces; however, as the speed sampling areas disabled by the study are completely nonfunctional, they decrease the flexibility of USVs during emergency collision avoidance, which limits the implementation of COLREGs Rule 2. Therefore, this paper proposes a DWA algorithm considering both the COLREGs rules and the influence of the wind and wave on ship navigation for the practical application of USVs. When scoring the sampled speeds, a series of speed-influencing factors, such as wind direction and angle, were considered to ensure that the improved speed calculation was closer to the actual speed for ship navigation. When a ship encounters two ships, the ship is given different priorities for port and starboard steering corresponding to positive and negative angular velocities for different avoidance situations to ensure that the collision avoidance behavior of the ship is more in line with the COLREGs.

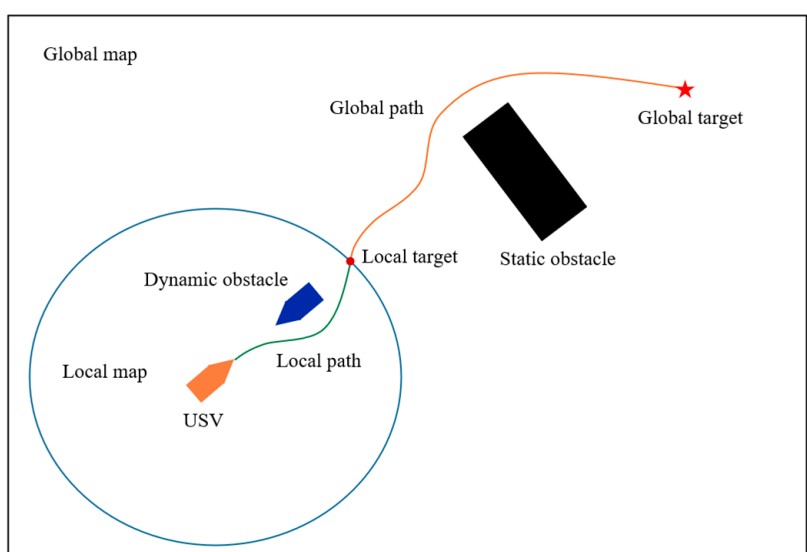

**Figure 1.** Global and local path planning combined with collision avoidance.

The remainder of this paper is organized as follows: Section 2 introduces the principle of the traditional DWA algorithm. Section 3 introduces an improved algorithm considering the COLREGs rules in windy and wavy environments. Section 4 simulates the proposed algorithm, demonstrating that the improved algorithm in this paper follows the COLREGs and possesses higher flexibility to make clever use of the wind and wave for collision avoidance and navigation. Section 5 discusses and analyzes the experimental results. Section 6 concludes the paper. Abbreviations appearing in this article can all be found in Table A1 in Appendix A.

## 2. DWA

DWA describes the obstacle avoidance problem as an optimization problem with constraints in the velocity space, which include the incomplete constraints of the USV, constraints of the environmental obstacles, and constraints of the dynamics of the USV. A schematic of the velocity vector space of the DWA algorithm is shown in Figure 2. The horizontal coordinate is the USV angular velocity, and the vertical coordinate is the USV linear velocity. The entire region is divided into safe and unsafe regions. In particular, the gray region is the safe region, the red rectangular box in the middle is the speed range that can be reached by considering the hardware constraints of the USV, and the remaining orange region, after excluding the unsafe region in the dark gray, is the final dynamic window.

The objective function of the standard DWA is:

$$G(v,\omega) = \alpha \times heading(v,\omega) + \beta \times distance(v,\omega) + \gamma \times velocity(v,\omega) \tag{1}$$

where $\alpha$, $\beta$, and $\gamma$ are weighting coefficients; $heading(v, \omega)$ is the azimuth evaluation function; $distance(v, \omega)$ is the obstacle distance evaluation function; and $velocity(v, \omega)$ is the speed evaluation function. $heading(v, \omega)$ is used to calculate the USV orientation score. During the movement of the USV, $heading(v, \omega)$ was used to drive the USV toward the target point. As $\Delta\theta$ decreases, the angle between the bow of USV and target point decreases. Meanwhile, the increase in the value of the $heading(v, \omega)$ function increases $G(v, \omega)$. The schematic of the DWA algorithm is shown in Figure 3.

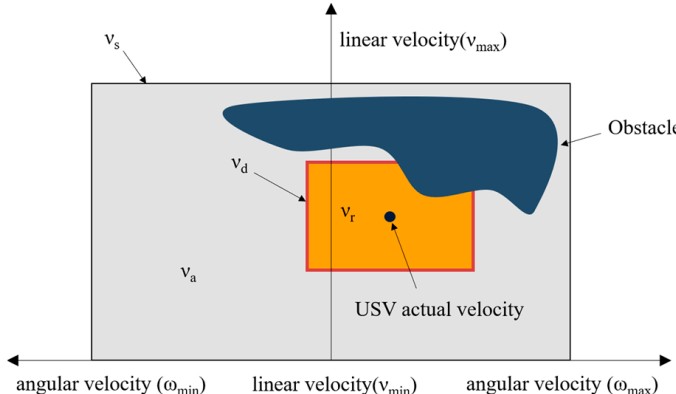

**Figure 2.** Schematic of the velocity vector space.

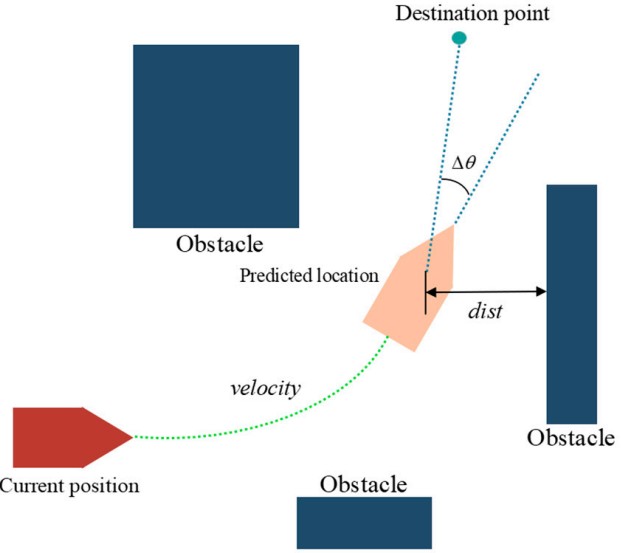

**Figure 3.** Schematic of the dynamic window approach (DWA) algorithm.

The obstacle distance evaluation function $distance(v, \omega)$ was used to calculate the USV score from the nearest obstacle, which reflects its obstacle avoidance ability. If the distance of the USV predicted trajectory from the obstacle is greater than the safe distance of the USV, there is no risk of collision. Conversely, as the distance of the nearest obstacle decreases, the $distance(v, \omega)$ function value decreases, the risk of collision increases, and the predicted trajectory is discarded.

The velocity evaluation function $velocity(v, \omega)$ is used to calculate the speed of the USV. In ensuring that no obstacles are encountered, the increases in the speed of $velocity(v, \omega)$ result in a greater value of the function and vice versa.

By adjusting these parameters, the focus of the USV movement pattern is regulated. For example, when the $\alpha$ value is high, the USV tends to use a speed group with the bow facing the target point, thereby appropriately reducing the obstacle distance and forward speed. When the $\beta$ value is high, the USV tends to select a speed group that is farther from the obstacle, the bow facing can be appropriately deviated from the target point, and the

forward speed can be appropriately reduced. At higher $\gamma$ values, USV favors the higher speed groups, while the effects of bow orientation and distance from the obstacles are appropriately attenuated.

The evaluation function is then normalized by

$$normal\_heading(i) = \frac{heading(i)}{\sum_{i=1}^{n} heading(i)} \tag{2}$$

$$normal\_distance(i) = \frac{distance(i)}{\sum_{i=1}^{n} distance(i)} \tag{3}$$

$$normal\_velocity(i) = \frac{velocity(i)}{\sum_{i=1}^{n} velocity(i)} \tag{4}$$

where $i$ is the $i$th trajectory, $n$ is all the trajectories sampled, $heading(i)$ is the azimuth function value obtained for the $i$th trajectory to be evaluated, $distance(i)$ is the obstacle distance function value obtained for the $i$th trajectory to be evaluated, and $velocity(i)$ is the velocity function value obtained for the $i$th trajectory to be evaluated; the above three function values are normalized accordingly. Finally, the motion attitude with the largest objective function value $G(v, \omega)$ is selected among all predicted trajectories as the optimal linear velocity $v$ and angular velocity $\omega$ for USV sampling. In this way, the DWA algorithm selects a path from the starting point to the target point and adjusts the path in real time according to the dynamic obstacles encountered to achieve collision avoidance.

## 3. Utility DWA (UDWA)

There are four main types of collision avoidance scenarios in COLREGs, namely head-on situation, crossing from the starboard side, crossing from the port side, and overtaking situation, as shown in Figure 4. The head-on case occurs when our ship and the target ship are moving head-on, that is, our ship and the target ship are traveling on an almost overlapping course. In this scenario, both our ship and the target ship should change their course to starboard to allow our ship to pass on the port side of the target ship (Rule 14), thereby ensuring safe navigation. The second and third scenarios are crossing situations, which are divided into the obstacle ships approaching from the starboard or port. When two ships are in a crossing situation, if the target ship is approaching from the starboard side of our ship, our ship should make way for the target ship and try to avoid crossing in front of the target ship (Rule 15), i.e., the second situation. When the target ship is approaching from the port side of our ship, our ship should maintain the original course and speed, i.e., the third case. The fourth scenario is an overtaking situation, where any overtaking ship shall avoid the ship being overtaken (Rule 13). If our ship is being overtaken, the original course and speed should be maintained.

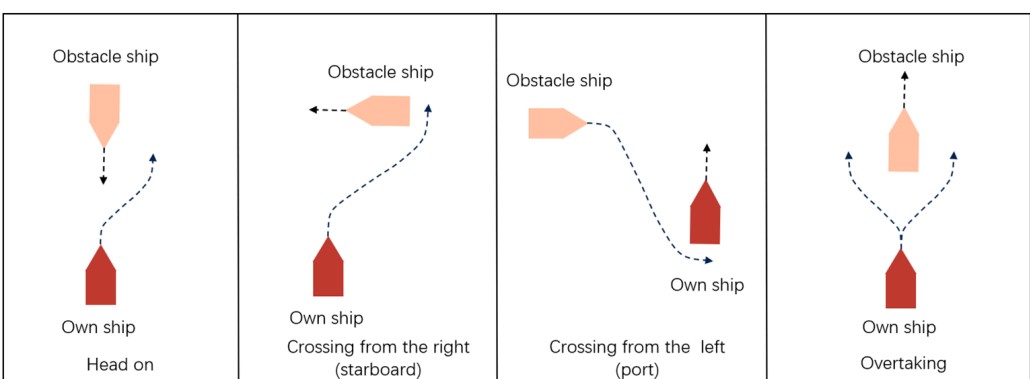

**Figure 4.** Main collision avoidance scenarios for ships in the COLREGs code.

For the above four main encounter avoidance scenarios, Ref. [22] proposed the COLREGs-compliant DWA (CCDWA) algorithm, which combines COLREGs based on the

DWA algorithm to increase its practicality. This study divides the dynamic window into port and starboard and removes the corresponding side from the search space for different encounter scenarios according to the COLREGs. The algorithm divides the velocity sampling space into two regions, as shown in Figure 5a, to disable different velocity sampling regions for different encounter scenarios.

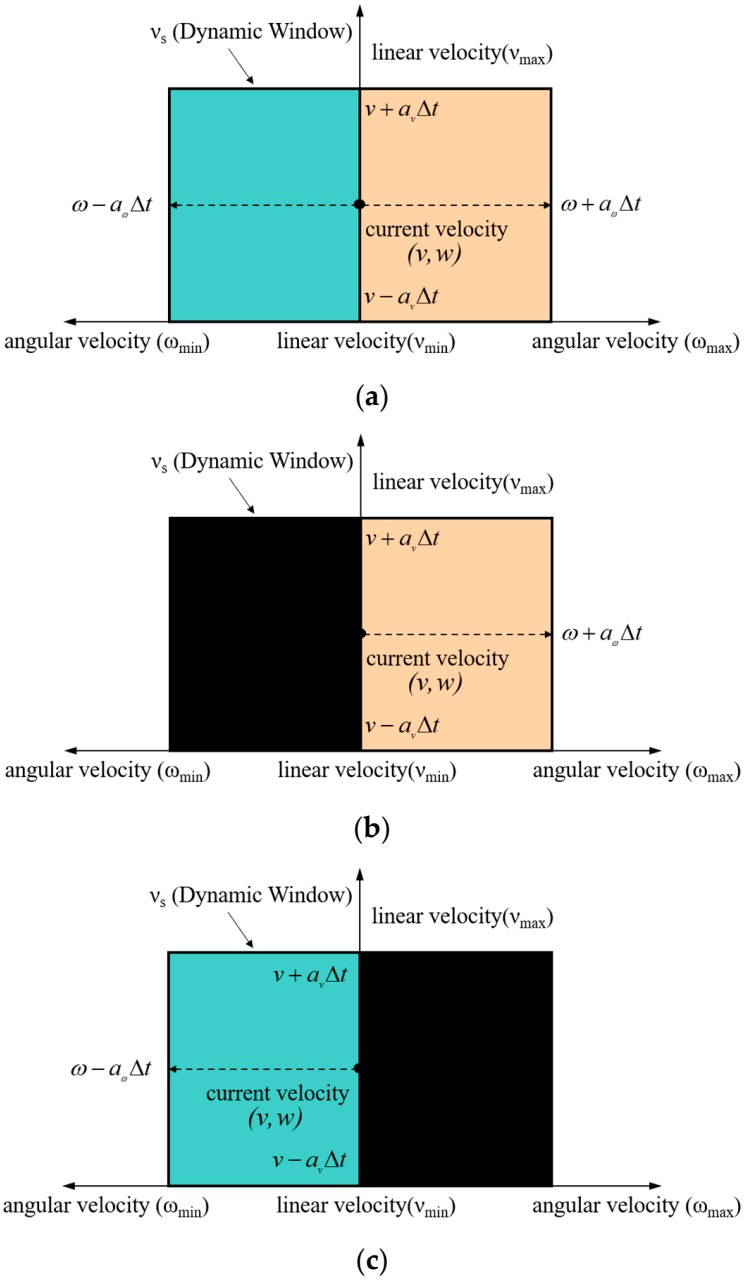

**Figure 5.** Schematic of the dynamic window of the CCDWA algorithm. (**a**) Schematic of the CCDWA algorithm speed sampling under normal sailing conditions. (**b**) Diagram of the area where sampling in the port direction is disabled. (**c**) Diagram of the area where sampling in the starboard direction is disabled.

According to the COLREGs rules, when a ship needs to turn to the starboard side, such as in the first and second scenarios mentioned above, the sampling area with a negative angular velocity is disabled, as shown in Figure 5b, and the black area is the nonsampling area. When the ship needs to turn to the port side, such as in the third case, the sampling area with the positive angular velocity is disabled, as shown in Figure 5c, and

the ship has to turn to the port side. However, according to COLREGs Rule 2, the ship can disobey the COLREGs rules when necessary to ensure navigational safety. As such, the above research CCDWA algorithm does not consider that the crew needs to make the most suitable collision avoidance decision in an emergency collision avoidance situation according to the encounter situation in which they are located. For example, when our ship is engaged in a head-on situation with the target ship and there is another ship or obstacle on the port side of the target ship, our ship will be in a dangerous situation and could even lead to a collision if it is still following the COLREGs rule, whereby our ship is steering toward starboard direction, as shown in Figure 6.

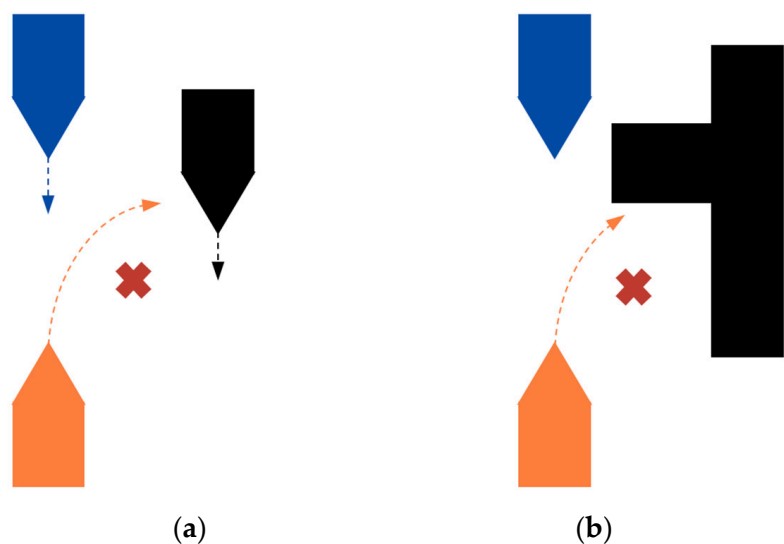

(**a**)  (**b**)

**Figure 6.** Examples of special ship encounters. (**a**) Presence of other dynamic obstacles on the starboard side of the vessel in head-on situations. (**b**) Presence of other static obstacles on the starboard side of the vessel in head-on situations.

As shown in Figure 6a, the target ship maintains its direction and speed, and there is a ship traveling head-on on the port side of the target ship. As such, if our ship steers toward the starboard side in accordance with the CCDWA algorithm, it may be very dangerous and can even lead to collision. As shown in Figure 6b, there is a stationary obstacle exactly on the port side of the target ship. If our ship steers toward the starboard side, it will be in danger or even result in collision. Therefore, the optimal choice for our ship in this situation should be to turn starboard as far as possible, and if the target ship has no intention of avoiding, our ship could quickly turn to the port side to avoid collision.

In this study, the problem is solved by assigning different priorities to two different velocity sampling regions for different encounter situations. The algorithm prioritizes starboard steering in general head-on situations. When starboard steering is not feasible, immediate port steering is performed, similar to crossing and overtaking situations. This ensures the flexible and feasible travel of our ship, which is more in line with COLREGs rules and safe navigation requirements. Figure 7 shows the sampling area (orange) with a higher priority, where the velocity group $(v, \omega)$ obtained is evaluated with a higher score than the lower priority sampling area (yellow area).

In order to differentiate from the concept of prohibited sampling areas in the CCDWA algorithm (black versus color in Figure 5), the UDWA algorithm uses the idea of different priorities (using orange versus yellow with different color depths, with darker color representing higher priority). In the face of emergency collision avoidance (e.g., Figure 6), the sampling method of CCDWA algorithm (shown in Figure 5) is too rigid which can easily cause danger or even collision, while the sampling method of UDWA algorithm (shown in Figure 7) will avoid the collision by turning to the port side, which is more flexible and less

dangerous compared to the sampling method shown in Figure 5, and Figure 7 has more flexibility and less danger.

When our ship and the target ship are under the head-on situation and the first crossing encounter and the overtaking scenario, i.e., the first, second, and fourth scenarios in Figure 4, our ship first considers steering to the starboard direction and gives priority to sampling the speed group $(v, \omega)$ with $\omega > 0$ (orange area). When there is an obstacle in the starboard direction or the other ship is not suitable for steering, the scores obtained will be lower than those of the speed group $(v, \omega)$ with $\omega < 0$ (yellow area), and the speed group of the yellow area is sampled, as shown in Figure 7a.

In the second scenario of the crossing situation or overtaking situation, i.e., the third and fourth scenarios in Figure 4, our ship prioritizes steering to the port direction and samples the speed group $(v, \omega)$ of $\omega < 0$ (orange area). When there is an obstacle in the port direction or other ships are unsuitable for steering, the scores will be lower, and the speed group $(v, \omega)$ of the yellow area will be sampled when its value is lower than the speed group $(v, \omega)$ of $\omega > 0$ (yellow area) to guarantee the flexibility of ship steering and optimum collision avoidance decision, according to the specific situation of the avoidance of collision, as shown in Figure 7b.

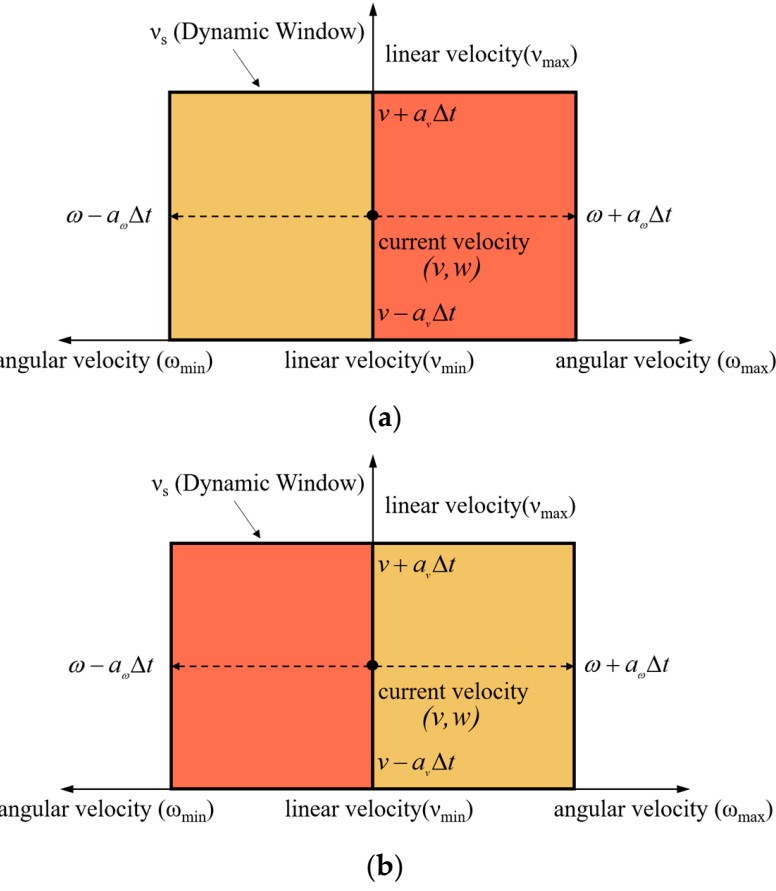

**Figure 7.** Schematic of the priority sampling by the UDWA. (**a**) Schematic of the priority sampling area in the starboard direction. (**b**) Schematic of the priority sampling area in the port direction.

Weather and wind–wave environment are not negligible factors in actual ship navigation. As such, it is important for a ship to navigate safely at a high sea level. The natural stalling of a ship's navigation is related to the maritime environment and the ship itself, with the effects of wind and waves being the most important environmental factors. However, there is no mature DWA algorithm to consider the complex weather and sailing environment. In this paper, the stall sailing formula of Liu Feng considers the size and direction of the wind and waves, the shipload, and other factors, which is more similar to

the ship simulation of an actual navigational environment. The formula is proposed on the basis of Aertssen's formula and the Scientific Research Institute of Maritime Transport of the USSR through the hull stall model fitting method by selecting about 150 sets of actual observational data covering various elements such as speed, wind field, waves, and so on. The statistical fitting test was carried out by combining the real ship data of the vessel "Longlin", and the test results of the real ship observation data show that the error between the predicted speed and the actual speed is not more than 1 knot. As shown in Equation (5):

$$V_a = V - \left(1.08 \times h - 0.126 \times q \times h + 2.77 \times 10^{-3} \times F \times \cos \varepsilon \right)\left(1 - 2.33 \times 10^{-7} \times \Delta \times V \right) \tag{5}$$

where $V_a$ is the actual ship speed (nautical miles per hour), $V$ is the speed of the ship in calm water (nautical miles per hour), $F$ is the actual wind speed (m/s), $h$ is the height of waves (m), $\Delta$ is the ship's deadweight (tons), $q$ is the angle of the wave direction to the sailing direction (radians), and $\varepsilon$ is the angle of the wind and waves to the ship's direction (radians).

The speed evaluation function $velocity(v, \omega)$ is modified to take only the absolute value of the speed in still water as the evaluation score. The highest speed in still water does not indicate the highest speed under the influence of sea breeze. The sample ship speed $(v, \omega)$ is brought into variable $V$. After calculation, the ship's sailing speed $V_a$ can be obtained under the actual wind and wavy environment and is scored; the larger the $V_a$ value is, the higher the actual speed of the ship's sampled speed group $(v, \omega)$ traveling under the influence of the sea wind is and the larger the score of the speed evaluation function $velocity(v, \omega)$ in the objective function is; thus, $(v, \omega)$ is selected as the traveling speed of the ship.

## 4. Simulation and Analysis

We compared the improved UDWA algorithm with the CCDWA algorithm to verify the effectiveness and feasibility of the algorithm. In this study, three simulations were designed. Section 4.1 presents the emergency collision avoidance of a ship completing three scenarios of head-on situation, crossing situation, and overtaking situation. Section 4.2 establishes the navigation task of a ship in waters near Ikaria island under windy and wavy conditions. Section 4.3 discusses the emergency collision avoidance of a ship in the waters near Ikaria Island under windy and wavy conditions. The algorithms were applied on a computer with the following configurations: Windows 64-bit with 12 cores (Intel(R) Core(TM) i7-12700F@2.10 GHz) and 16 GB RAM, running with MATLAB R2022a implementation.

### 4.1. Improved Experiments Based on COLREGs Rule 2

Figure 8 shows the simulation of the emergency collision avoidance scenario of the CCDWA algorithm for the head-on situation; the red ship in the figure is our USV, the blue ship is the head-on USV, the black ship is the other USV causing the emergency collision avoidance scenario, and the dashed ellipse area is the ship domain of the corresponding ship, which is seven times the length of ship on the long axis and thrice the length of ship on the short axis.

As the CCDWA algorithm only samples the velocity in the starboard region ($\omega > 0$), while the velocity in the port region ($\omega < 0$) is excluded for the head-on situation, the ship will only steer to the starboard side to avoid collision. However, when there is another ship on the starboard side (black ship), as shown in Figure 8b, the ship may not adjust its attitude in time due to the limitation of its own motion model and then invade the field of the other ship, prompting danger or even collision. Although it can completely avoid collision, a great risk remains.

Figure 9 shows the emergency collision avoidance simulation of the improved algorithm proposed in this paper in the head-on situation. The improved algorithm prioritizes the sampling of the speed of the starboard region ($\omega > 0$). When there are other ships in the starboard direction, which is not conducive to steering, the speed of the port region ($\omega < 0$) will be sampled to achieve collision avoidance, as shown in Figure 9b. The passage

from the direction of the risk can greatly reduce the intrusion into the field of the other ships and the probability of collision.

Figure 10 shows the comparison of the distance between our ship and the nearest obstacle ship using the CCDWA and UDWA algorithms. As shown in the figure, the initial distance between the ships is equal, and the distance decreases until 12 s when the USVs are traveling toward the encounter scenario. As such, the collision avoidance behavior starts at 12 s, where the closest distance between the USVs using the CCDWA algorithm is 33.56 m, and the average distance is 54.80 m, whereas the closest distance between the USVs using the UDWA algorithm is 59.48 m, and the average distance is 78.50 m. As such, the collision avoidance ended, our ship moved from the remaining obstacle ship, and the distance increased.

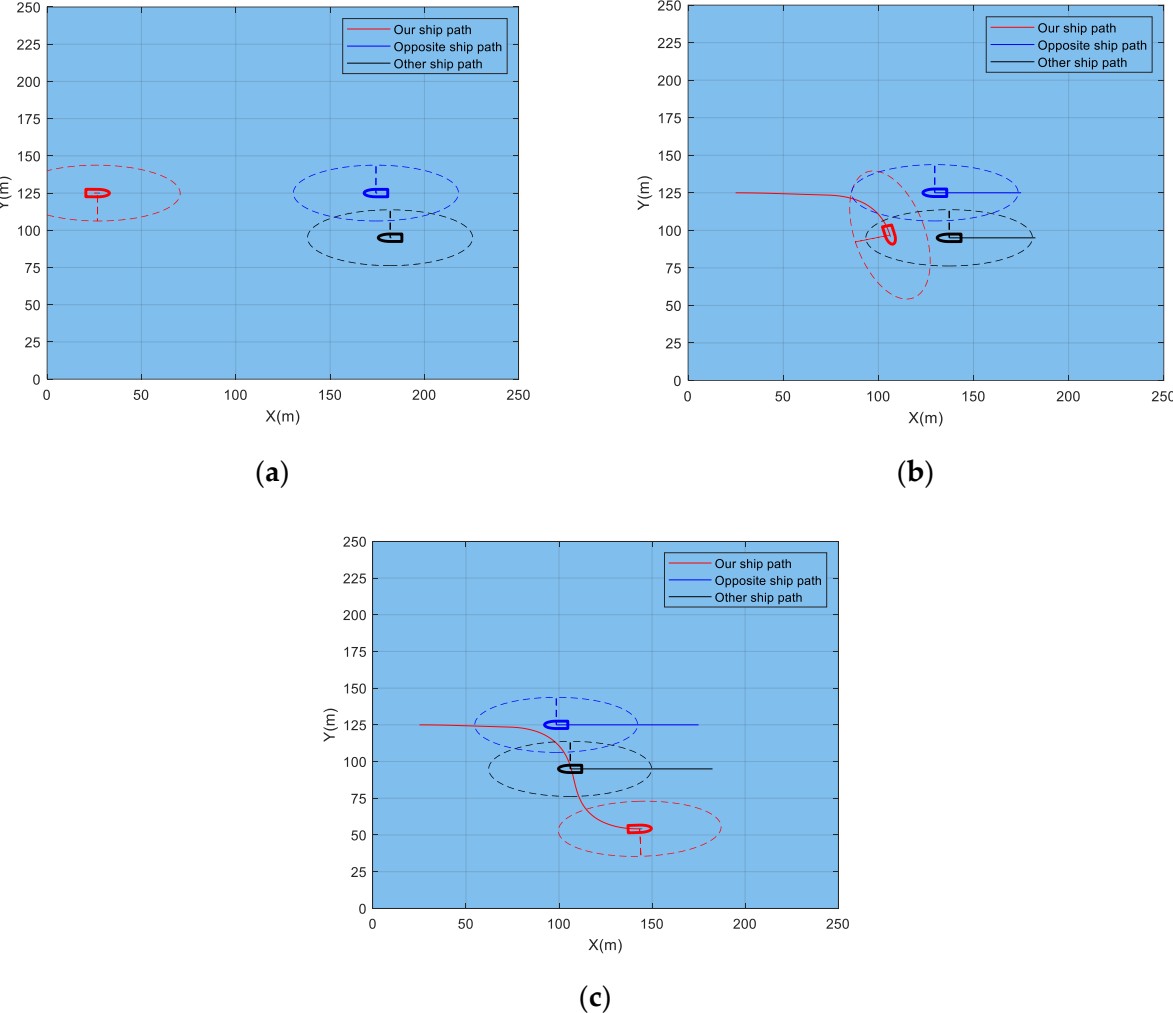

**Figure 8.** CCDWA algorithm for emergency collision avoidance in the head-on scenario. (**a**) Starting position of collision avoidance. (**b**) Start avoiding collisions. (**c**) End of collision avoidance.

Figure 11 shows the simulation of the CCDWA algorithm in the crossing situation, according to COLREGs Rule 15: when the other ship comes from the starboard side of our ship, it should pass from the starboard direction and try to avoid crossing the front of the ship. The CCDWA algorithm only samples the speed of the starboard area ($\omega > 0$), whereas that of the port area ($\omega < 0$) is excluded, which results in the ship passing through the starboard side only. As the target ship is followed by another ship, it is difficult to dodge when our ship is moving faster, thereby invading the ship domain of the rest of the ships and causing danger, as shown in Figure 11b. Figure 11c shows that the collision avoidance trajectory at 80–120 m was also zigzagged, and there was an emergency turning

behavior of the red USV even after the collision avoidance was completed to avoid the black USV at $x \in [80, 120]$, which is very dangerous for the large ship.

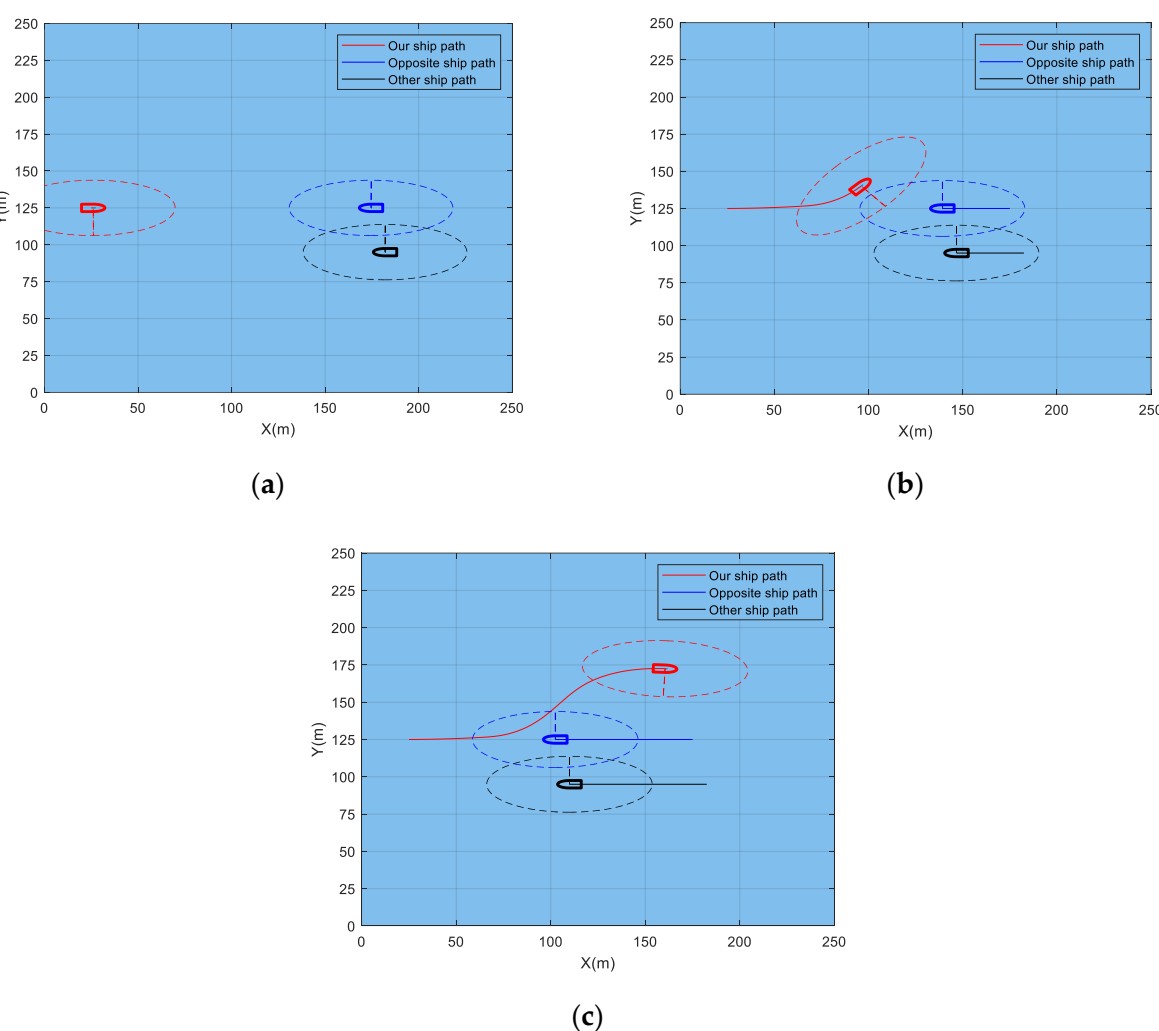

(**a**)　　　　　(**b**)

(**c**)

**Figure 9.** UDWA algorithm for emergency collision avoidance in the head-on scenario. (**a**) Starting position of collision avoidance. (**b**) Start avoiding collisions. (**c**) End of collision avoidance.

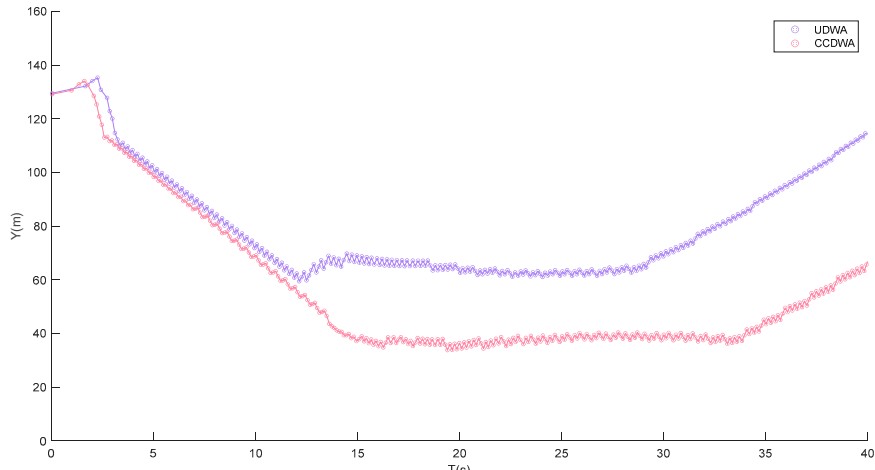

**Figure 10.** Comparison of the USV and nearest-ship distances obtained by the CCDWA and UDWA algorithms for the head-on scenario.

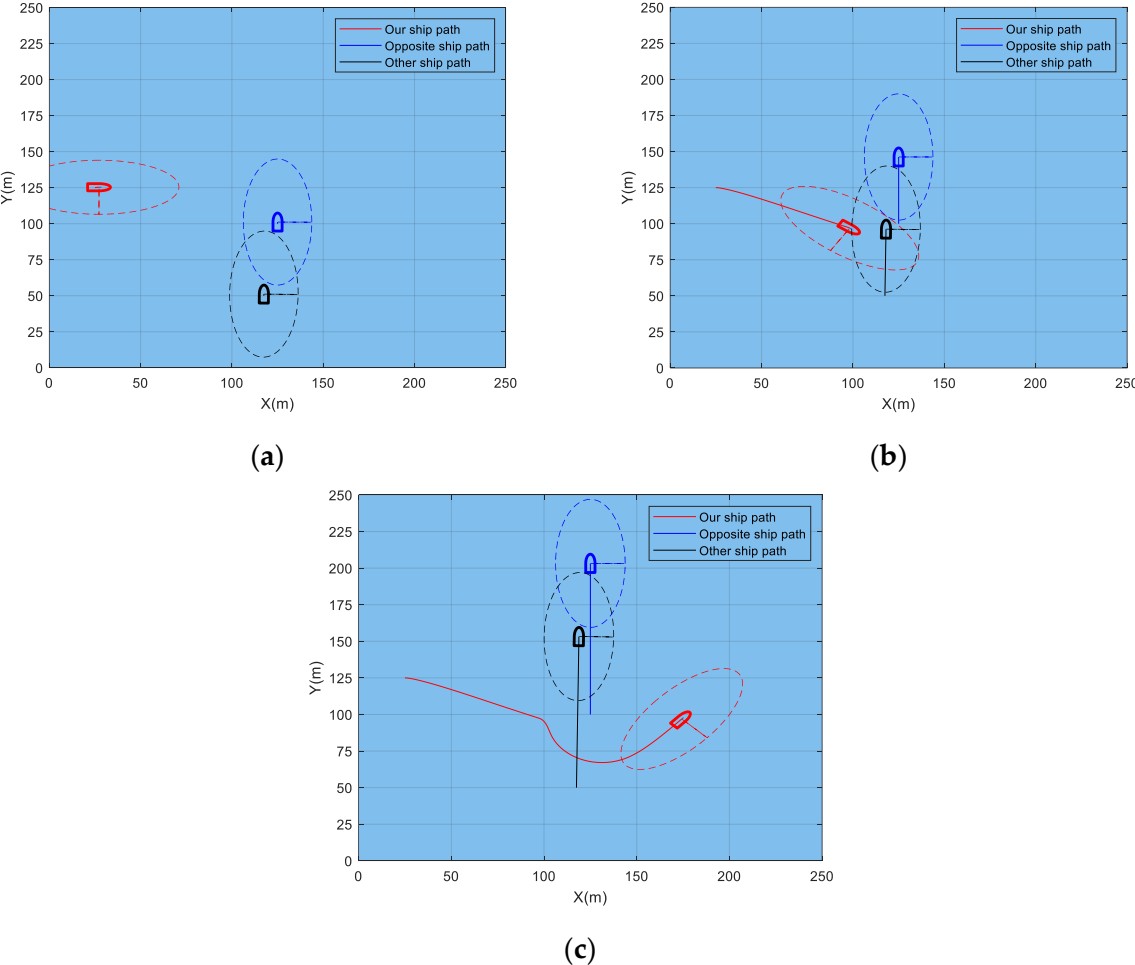

**Figure 11.** CCDWA algorithm for emergency collision avoidance in the crossing situation. (**a**) Starting position of collision avoidance. (**b**) Start avoiding collisions. (**c**) End of collision avoidance.

The improved algorithm prioritizes sampling the starboard area speed ($\omega > 0$) when there are other ship obstacles in the starboard direction and switches to sampling the port area speed ($\omega < 0$) to choose the less risky USV to pass in the port direction. According to COLREGs Rule 2, ships can disobey the COLREGs rules when necessary to ensure navigation safety. Thus, passing on the port side can be less risky in Figure 12b, thereby completing the collision avoidance and gradually returning to the established course.

As shown in Figure 13, the distance between the ships gradually decreases before 5 s. Using the CCDWA algorithm, the black ship that can only be avoided through the starboard direction reduces the timeliness of our ship's reaction, decreasing the distance to the minimum value of 18.16 m and average value of 50.93 m. In contrast, using the UDWA algorithm, the avoidance of collision from the port side to avoid the influence of the black ship allows our ship and the obstacle ship to always maintain a large distance with a minimum value of 60.73 m and average value of 66.90 m. Subsequently, the collision avoidance is ended, and the distance between the ships is gradually increased.

Figure 14 shows the emergency collision avoidance operation of the CCDWA algorithm in the overtaking situations. The port-side speed region ($\omega < 0$) is excluded; that is, the ship can only steer to the starboard side. After the pursuit of the blue ship, the red ship is forced to continue to steer on the starboard side due to the presence of the black ship, as shown in Figure 14b. Owing to the limitation of the motion model, consecutive steering on the starboard side allows the red ship to easily enter the domain of the other ships, as shown in Figure 14b. Although successful collision avoidance could be achieved, the extremely small distance between the ships poses a risk.

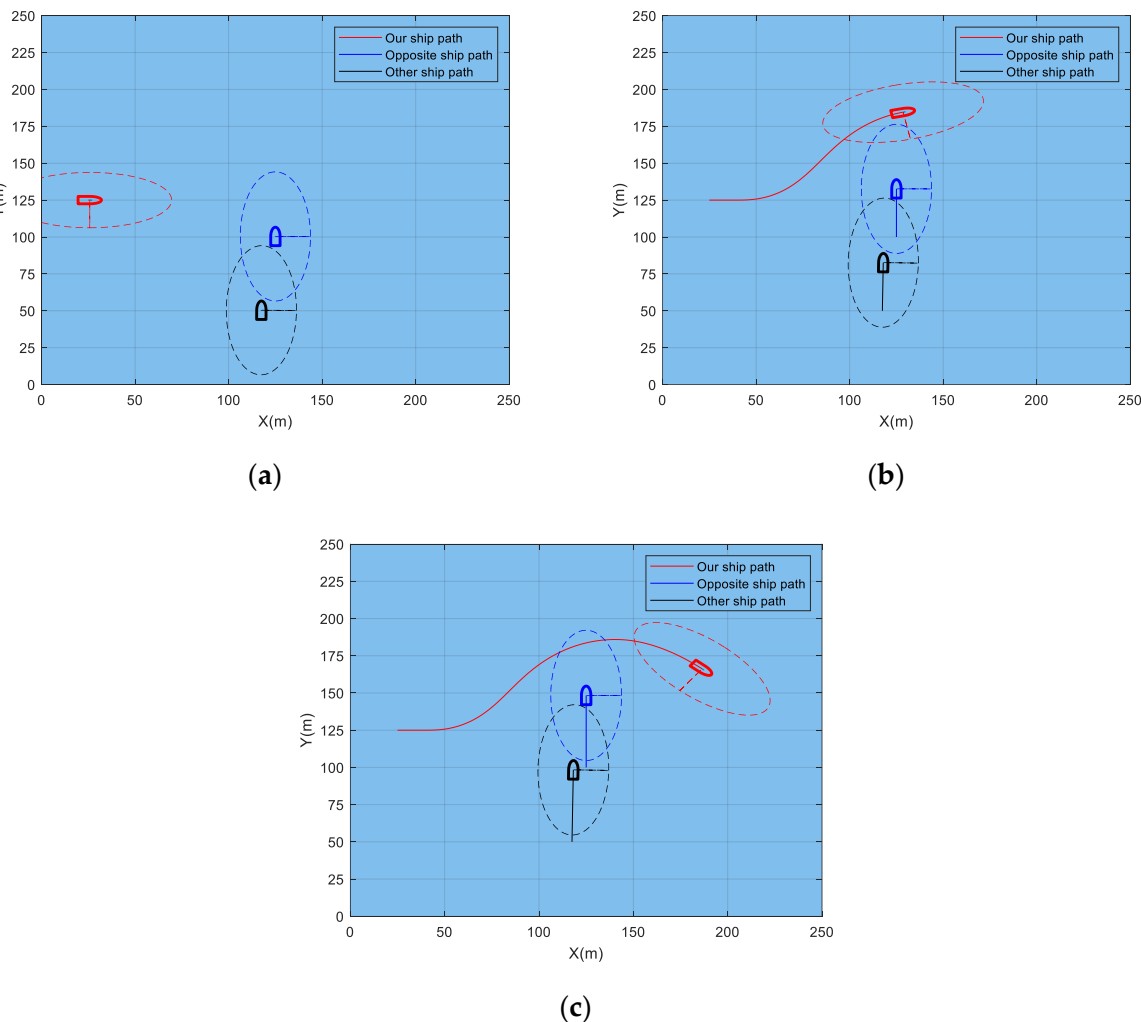

**Figure 12.** UDWA algorithm for emergency collision avoidance in the crossing situation. (**a**) Starting position of collision avoidance. (**b**) Start avoiding collisions. (**c**) End of collision avoidance.

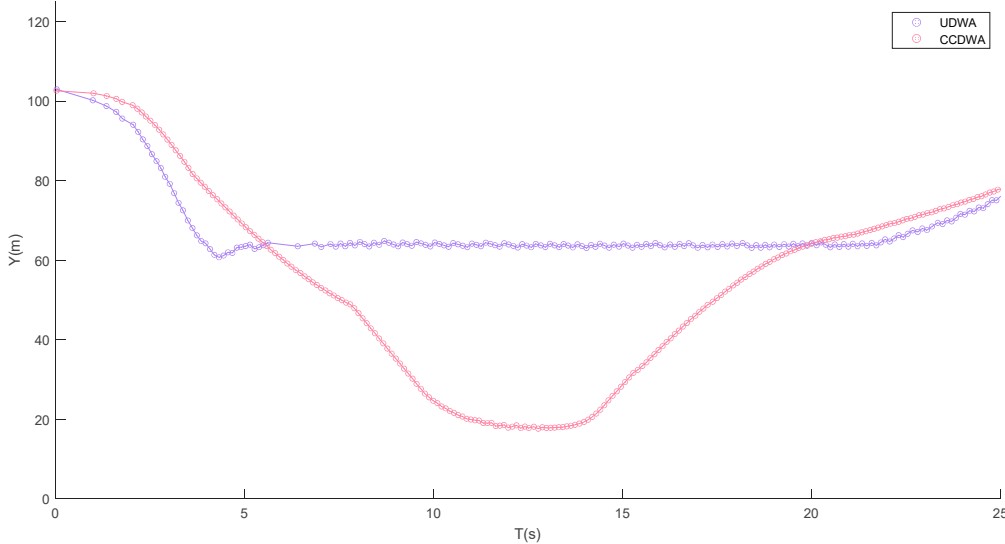

**Figure 13.** Comparison of USV and nearest-ship distances under the CCDWA and UDWA algorithms (crossing situations).

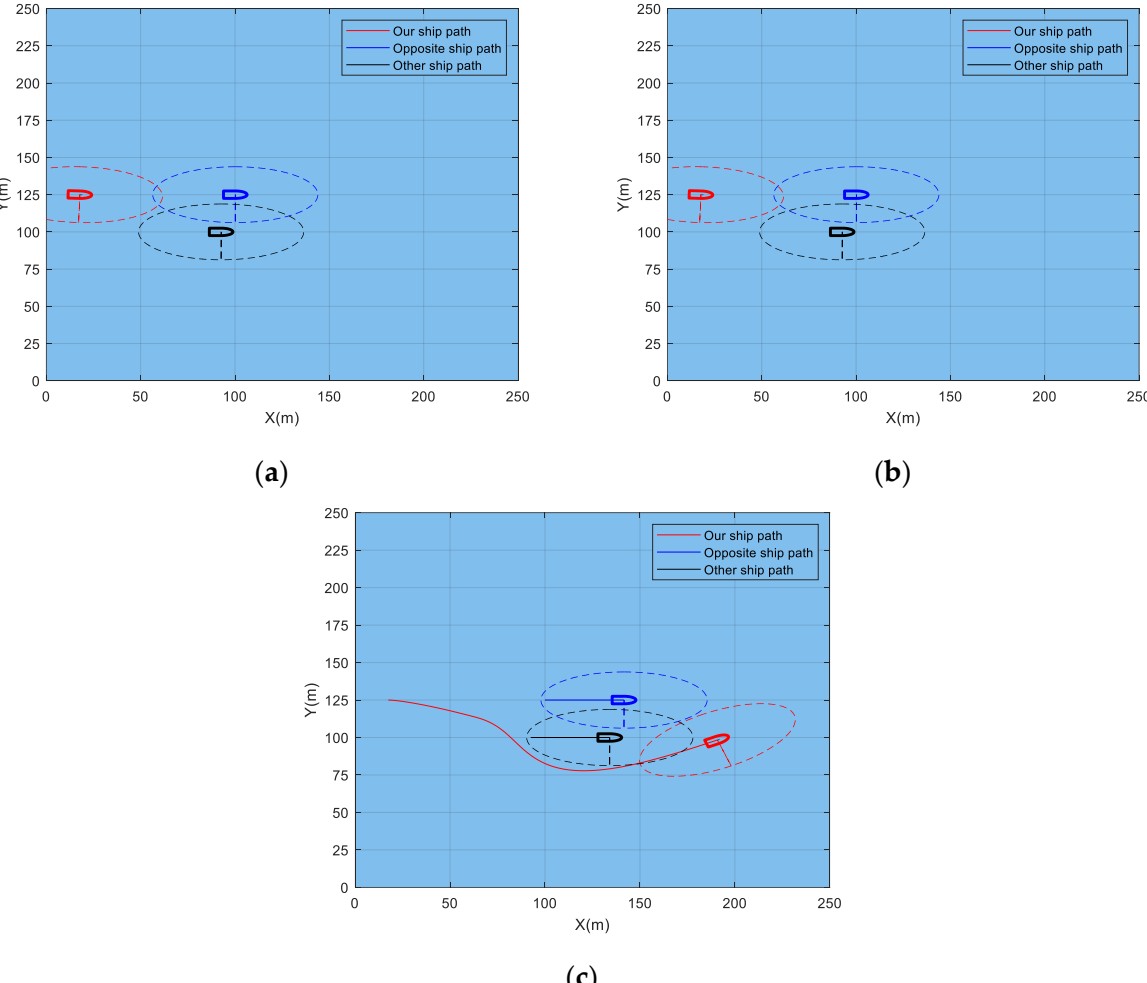

**Figure 14.** CCDWA algorithm for emergency collision avoidance in overtaking situations. (**a**) Starting position of collision avoidance. (**b**) Start avoiding collisions. (**c**) End of collision avoidance.

The improved algorithm proposed in this paper encounters an emergency collision avoidance scenario in overtaking situations. In particular, the objective function value decreases when pre-steering to the starboard direction due to the presence of other obstacles. As such, the velocity group of the port velocity region ($\omega < 0$) is sampled, and collision avoidance behavior is completed by steering to the port side to reduce the risk of collision, as shown in Figure 15b.

The improved DWA algorithm steers from the starboard side according to COLREGs Rules 13–15 in traditional collision avoidance scenarios and steers from the port side to avoid collision if the risk of steering from the starboard side is higher when facing emergency collision avoidance scenarios, thereby satisfying the provisions of COLREGs Rule 2, whereby the ship can avoid collision by not following the COLREGs rules when necessary. The improved DWA algorithm follows the COLREGs rules better and has better flexibility than the CCDWA algorithm in limiting the sampling area, which chooses the steering direction according to specific situations of the collision avoidance scenario, thereby realizing collision avoidance.

As shown in Figure 16, the speed of our ship is higher in the overtaking situation, and the distance with the obstacle ship gradually decreases. Subsequently, we obtain the collision avoidance behavior with a minimum distance of 19.13 m and an average distance of 37.93 m using the CCDWA algorithm and a minimum distance of 40.40 m and an average distance of 63.65 m using the UDWA algorithm. Our ship gradually moves from the obstacle ship after the collision avoidance operations are finished.

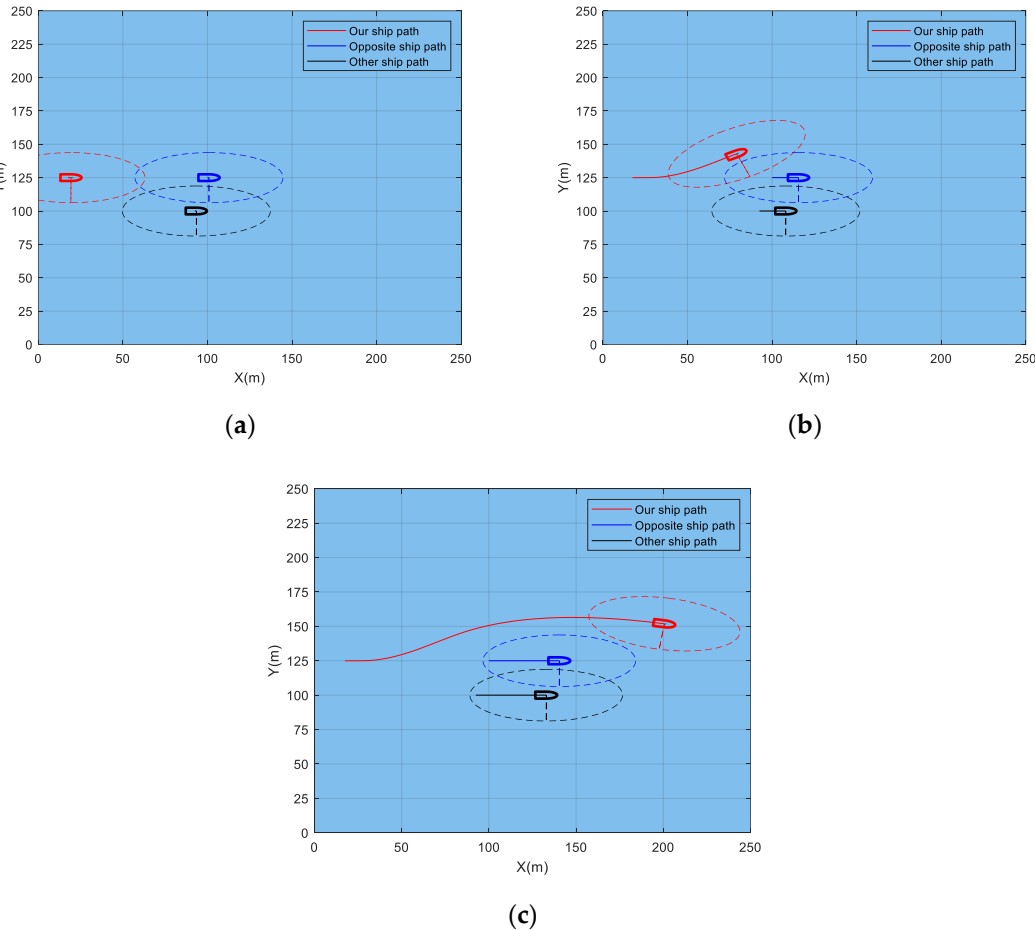

**Figure 15.** UDWA algorithm for emergency collision avoidance in overtaking situations. (**a**) Starting position of collision avoidance. (**b**) Start avoiding collisions. (**c**) End of collision avoidance.

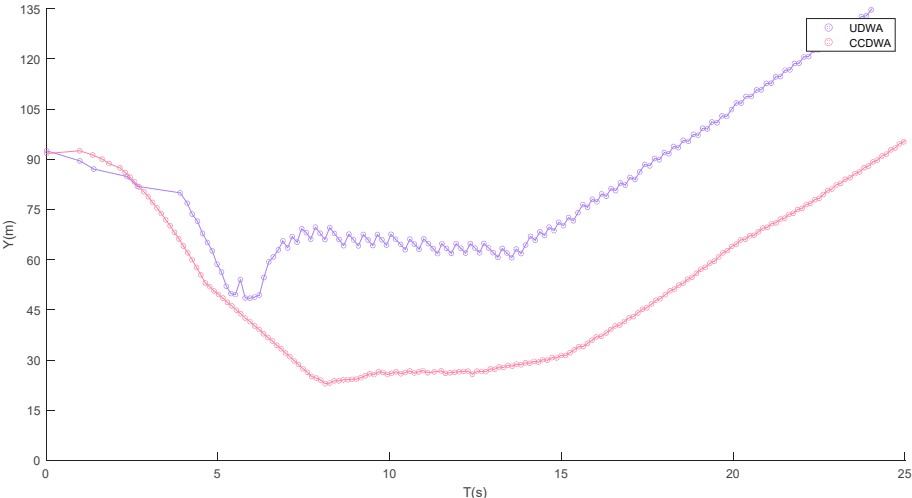

**Figure 16.** Comparison of the USV and nearest-ship distances under the CCDWA and UDWA algorithms (overtaking situations).

The distance comparisons in the three collision avoidance scenarios using the two algorithms are shown in Table 1. Comparing the CCDWA and UDWA algorithms, the distance between our ship and the obstacle ship can be farther, which reduces the risk of collision. The optimization rates in the head-on situation, crossing situation, and overtaking situation are 43.25%, 31.36%, and 67.81%, respectively.

**Table 1.** Distance between our ship and the obstacle ship in the collision avoidance scenario obtained by the CCDWA and UDWA algorithms.

| Encounter Scenario | Head-on | Crossing | Overtaking |
|---|---|---|---|
| closest distance (CCDWA) | 33.56 m | 18.16 m | 19.13 m |
| closest distance (UDWA) | 59.48 m | 60.73 m | 40.40 m |
| average distance (CCDWA) | 54.80 m | 50.93 m | 37.93 m |
| average distance (UDWA) | 78.50 m | 66.90 m | 63.65 m |
| optimization rate | 43.25% | 31.36% | 67.81% |

*4.2. UDWA Algorithm Validation Experiments in a Wind and Wave Environment*

For the validation of the algorithm for navigating in windy and wavy environments, the waters around the Ikaria island were used in the simulation environment, as shown in Figure 17. Two sea-breeze conditions are simulated in this study to verify the effectiveness of navigating in windy and wavy environments, namely a higher wind speed in the south and lower wind speeds in the north, as shown in Figure 18a, and a higher wind speed in the north and lower wind speed in the south, as shown in Figure 18b. The wind and waves move faster in the warm-colored areas and slower in the cool-colored areas. The red arrows show the direction of the wind and waves.

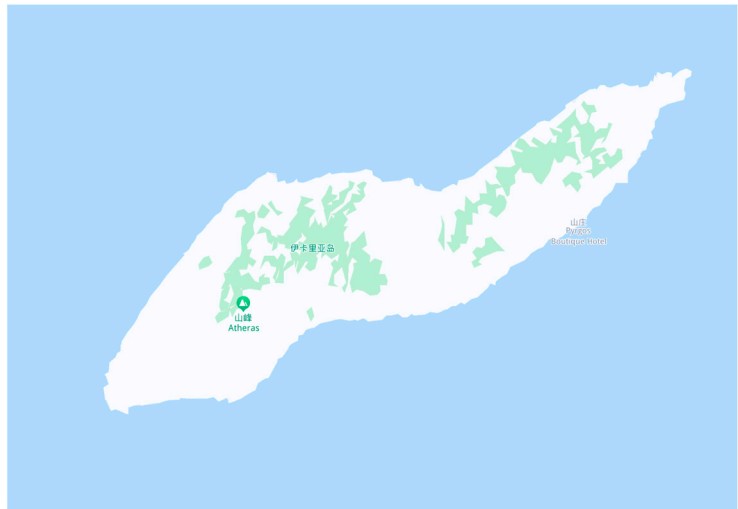

**Figure 17.** Map of the waters of Ikaria Island.

In this experiment, the map was first rasterized. Each square in the rasterized map has four attributes: the identification of whether it is an obstacle or not, the wind direction $W_\theta$, wind size $F$, and wave height $h$. The simulated wind speed ranges from 0 to 7 m/s, with the corresponding wave height shown in Table 2. The USV motion parameter settings are shown in Table 3.

USV sailing to different grids has different wind magnitude $F$ and direction $W_\theta$ and wave height $h$. Substituting them into Equation (5), the actual and optimal sailing speeds $V_a$ are obtained according to the best actual sailing speed. First, the two USVs were positioned in the southwest waters, and the target point was set in the northeast waters. Two different windy and wavy environments were imposed to observe the navigation of the USVs. The red USV uses the UDWA algorithm, and the sailing route is represented by a red line, whereas the blue USV uses the traditional DWA algorithm, and the sailing route is represented by a blue line. The red "*" symbol represents the target point. First, the windy

and wavy environment with higher wind speed in the south in Figure 18a was used; the sailing results are shown in Figure 19.

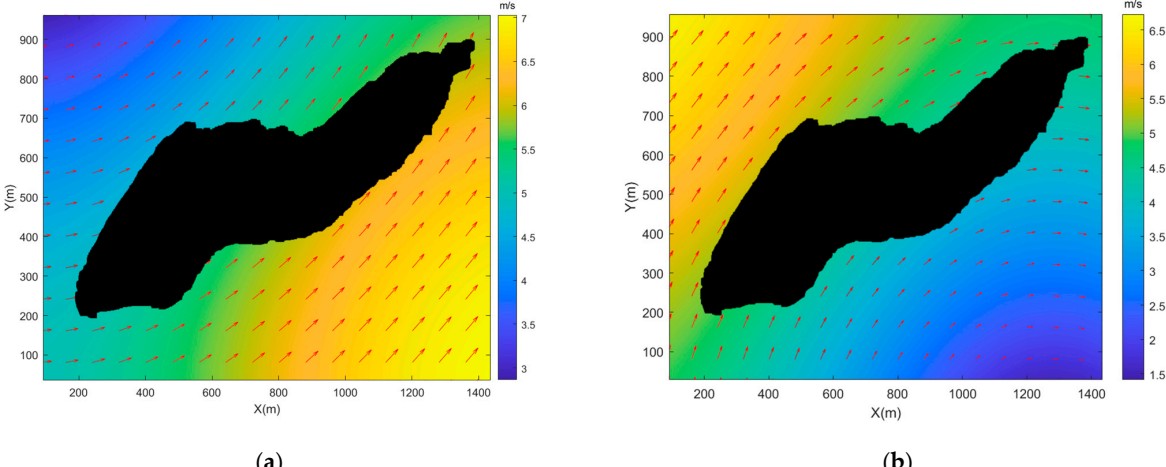

(**a**)                                                                                                              (**b**)

**Figure 18.** Map of the sea-breeze environment on the waters of Ikaria Island. (**a**) Wind and wave diagrams with fast winds in the south and slow winds in the north. (**b**) Wind and wave diagrams with fast winds in the north and slow winds in the south.

**Table 2.** Wind magnitude and wave heights.

| Wind Rating | Wind Size *F* (m/s) | Wave Height *h* (m) |
| --- | --- | --- |
| 0 | 0–0.2 | 0.0 |
| 1 | 0.3–1.5 | 0.1 |
| 2 | 1.6–3.3 | 0.2 |
| 3 | 3.4–5.4 | 0.6 |
| 4 | 5.5–7.0 | 1.0 |

**Table 3.** USV motion parameters.

| Parameters | Value |
| --- | --- |
| Deadweight of the ship | 20 t |
| Maximum movement speed | 7 m/s |
| Minimum movement speed | 0 m/s |
| maximum acceleration | 2 m/s$^2$ |
| Maximum angular velocity | $\pm$20 deg/s |
| Maximum angular acceleration | $\pm$5 deg/s$^2$ |
| $\alpha$ | 0.2 |
| $\beta$ | 0.5 |
| $\gamma$ | 0.3 |

As can be seen in Figure 19, the red USV has a higher sailing speed and arrives at the target point at 52 min in the windy and wavy environment with higher wind speed in the north, whereas the blue USV does not account for the windy and wavey environment and sails slower and arrives at the target point at 65 min. Figure 20 shows the experimental results obtained from the simulation using the environment with a higher windy and wavy environment in the north, as shown in Figure 18b. As shown in Figure 20, using the wind and wave field with higher wind speeds in the north, the red USV will choose the northern waters with higher wind speeds to navigate safely and arrive at the target point in 48 min owing to its higher wind speeds than the south water, whereas the blue USV takes 67 min to arrive at the target point. After the experiments under two wind and wave fields, the red ship chooses to travel different routes. Moreover, objective function Equation (1) considers the wind and wave direction and size factors when calculating $velocity(v, \omega)$ and

chooses the sampling speed group $(v, \omega)$ with a higher actual sailing speed, whereas the blue ship using the traditional DWA algorithm does not have the ability to choose optimal sailing routes.

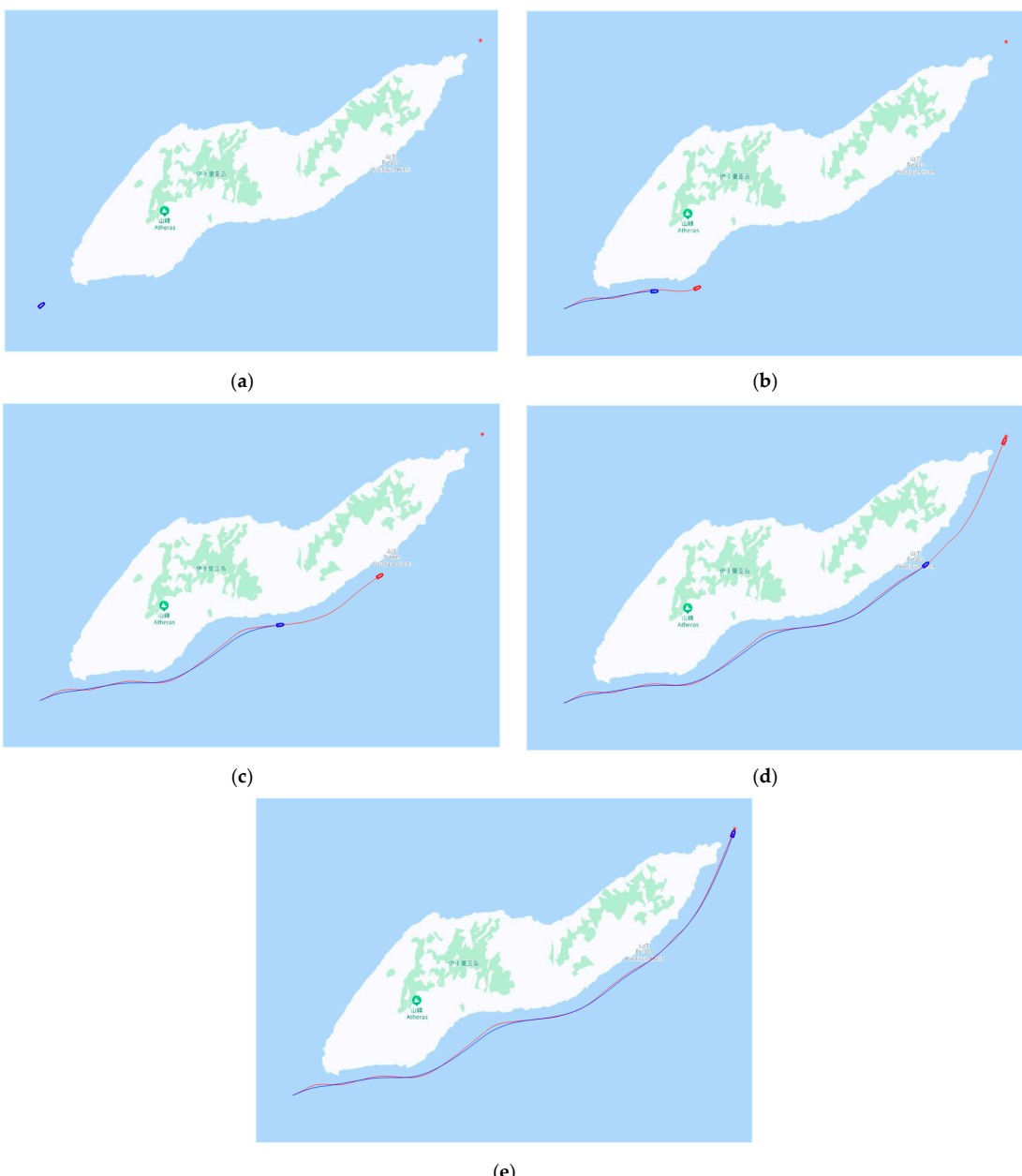

**Figure 19.** Simulation of the sailing process of USVs in a southern wind field with higher wind speeds. (**a**) 0 min. (**b**) 13 min. (**c**) 31 min. (**d**) 48 min. (**e**) 67 min.

*4.3. Validation Experiment of the Collision Avoidance Based on the COLREGs Rules under the Effects of the Wind and Wave*

Combining the improved UDWA algorithm that considers the COLREGs rules with the navigation in windy and wavy environments is proposed in this paper, and a practical DWA unmanned ship collision avoidance algorithm that follows the COLREGs rules in windy and wavy environments was demonstrated. Figure 21 shows an experimental simulation that employs the windy and wavy environments shown in Figure 18a to validate the sufficient flexibility of the USVs to follow the COLREGs Rule 2.

**Figure 20.** Simulation of the sailing process of the USVs in a northern wind field with higher wind speeds. (**a**) 0 min. (**b**) 13 min. (**c**) 39 min. (**d**) 52 min. (**e**) 65 min.

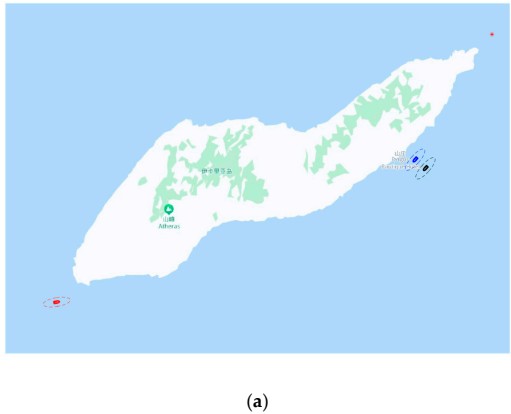

(**a**)

**Figure 21.** *Cont.*

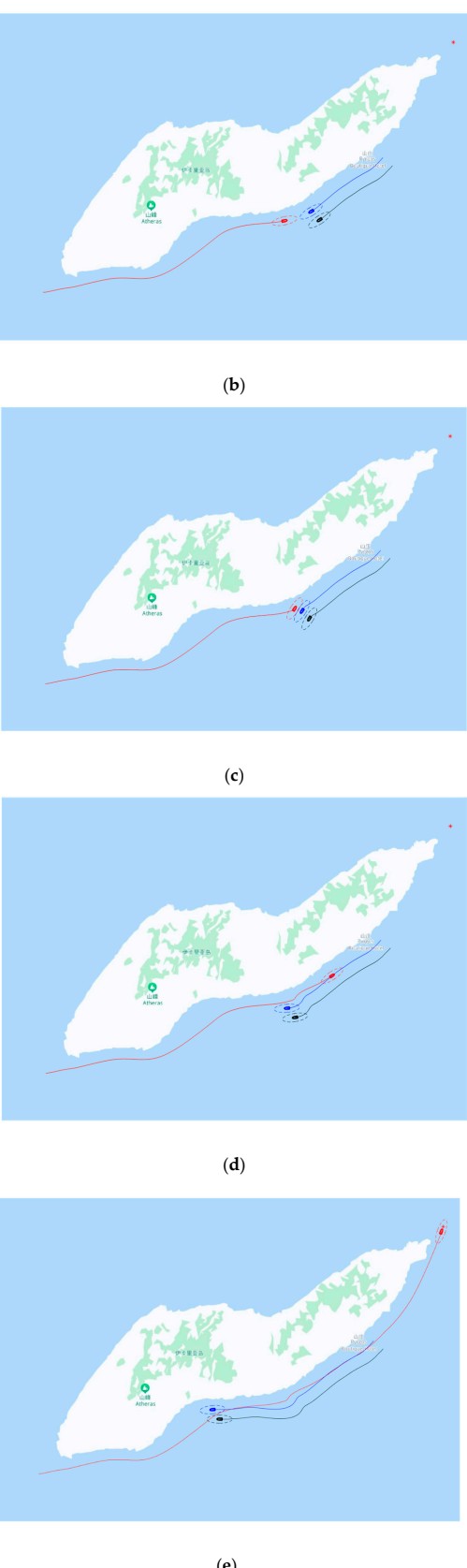

**Figure 21.** Validation of the UDWA algorithm considering the COLREGs rules for emergency collision avoidance in windy and wavy environments. (**a**) USV initial position. (**b**) USV begins emergency collision avoidance. (**c**) USV is in emergency collision avoidance. (**d**) End of USV emergency collision avoidance. (**e**) Completion of USV cruise.

As shown in Figure 21b–e, the USV can flexibly follow the COLREGs Rule 2 when facing emergency collision avoidance in the encounter scenario, whereby it should not follow the COLREGs to navigate safely and avoid collision, if necessary. In this case, collision avoidance can be accomplished by steering to the port side. Moreover, in the windy and wavy environment shown in Figure 18a, the USV traveled on the south side of the water to verify the validity of the USV considering the windy and wavy environment.

## 5. Discussion

It can be concluded from Section 4.1 that the optimization rates in the head-on situation, crossing situation, and overtaking situation are 43.25%, 31.36%, and 67.81%, respectively. In emergency collision avoidance scenarios, it is clear that changing the collision avoidance heading has a better effect than just increasing the weight of the distance function, which not only satisfies the safety requirements, but also meets the requirements of COLREGs Rule 2. As can be seen from Experiment 4.2, the UDWA algorithm that takes wind direction into account allows the ship to reach the target point faster compared to the DWA algorithm, and to change the sailing direction at any time with the different wind directions. This way, not only can we get a faster moving speed, but also save fuel to a certain extent and achieve the purpose of saving fuel. In the experiments in Section 4.3, the UDWA algorithm can still safely complete collision avoidance and reach the target point in the presence of wind and waves. The algorithm satisfies not only the security requirements but also the economic requirements. The experimental background set in this paper is a simple emergency collision avoidance scenario for three ships, all of which have achieved relatively good results; however, in the actual sailing process of emergency collision avoidance of a wide range of situations and complexity, the DWA algorithm seems too rigid, while the UDWA can better implement the COLREGs rules to complete the collision avoidance.

## 6. Conclusions

USVs must follow the COLREGs rules to avoid collision and navigate safely. Few existing DWA algorithms consider the COLREGs rule, including the CCDWA algorithm; however, the algorithm has insufficient flexibility, and it is too rigid during emergency collision avoidance situations. Moreover, windy and wavy environments are important factors affecting the navigation of USVs. In particular, the clever use of wind direction and wind speed can achieve the target point faster and save fuel. In this study, the limitation of the sampling area was first changed to include prioritizing the sampling area to improve the flexibility of the CCDWA algorithm. Different priorities were given to the port and starboard speed sampling areas in the face of different collision avoidance scenarios. Subsequently, the original speed function was replaced by an improved speed formula that considers a windy and wavy environment, thereby allowing the USV to cleverly use the wind speed, wind direction, and waves to avoid collision and save time. Two parts of the improvement were simulated separately. The simulation results showed that the UDWA algorithm optimized the distance to the obstacle ship by 43.25%, 31.36%, and 67.81% in a head-on situation, crossing situation, and overtaking situation, respectively, compared to the CCDWA algorithm. The improved algorithm can better follow the COLREGs rules and has better flexibility when facing emergency collision avoidance situations. Moreover, the USV can navigate in windy and wavy environments with full consideration of the wind and wave directions and sizes. Finally, two parts of the improved algorithm were combined with simulation. The experimental results showed that the improved algorithm can consider windy and wavy environments and follow the COLREGs rules for more flexible collision avoidance.

Although the research in this paper has considered the influence of the navigation environment on the actual speed of the ship, there are still some deficiencies. The factors affecting the actual speed of a ship are numerous and complex; in addition to the factors mentioned in this paper, the structure and shape of the ship are also one of the factors affecting the actual speed. Moreover, since the DWA algorithm is a local path planning

algorithm, the paths obtained on large-scale maps may not be the optimal paths, and the combination of the improved global path planning algorithm may achieve better results. The deficiencies pointed out above are also the objectives of the work to be solved in future research.

**Author Contributions:** Conceptualization, X.Y. and H.W.; methodology, X.Y.; software, X.Y.; validation, X.Y., H.W. and G.H.; formal analysis, X.Y.; investigation, X.Y.; resources, X.Y.; data curation, C.T. and G.H.; writing—original draft preparation, X.Y.; writing—review and editing, X.Y.; visualization, X.Y.; supervision, H.W.; project administration, H.W.; funding acquisition, H.W. All authors have read and agreed to the published version of the manuscript.

**Funding:** This research was funded by National Natural Science Foundation of China, grant number U1964202.

**Institutional Review Board Statement:** Not applicable for studies not involving humans or animals.

**Informed Consent Statement:** Not applicable for studies not involving humans.

**Data Availability Statement:** Data sharing not applicable. No new data were created or analyzed in this study. Data sharing is not applicable to this article.

**Acknowledgments:** The authors are grateful to the anonymous reviewers for their valuable comments and suggestions that helped improve the quality of this manuscript.

**Conflicts of Interest:** The authors declare no conflict of interest.

## Appendix A

**Table A1.** List of abbreviations.

| Abbreviation | Meaning |
| --- | --- |
| ACO | ant colony optimization |
| APF | artificial potential field |
| CCDWA | COLREGs-compliant DWA |
| COLREGs | International Regulations for Preventing Collisions at Sea |
| DVOI | degree of velocity obstacle intrusion |
| DWA | dynamic window approach |
| GA | genetic algorithm |
| MPDP | the multi-objective peak DP |
| NN | neural networks |
| RRT | rapidly exploring random tree |
| TVOI | time of velocity obstacle intrusion |
| UDWA | utility DWA |
| USV | unmanned surface vessels |
| WSO | white shark optimizer |

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
