# Peer review of "Unmanned Vessel Collision Avoidance Algorithm by Dynamic Window Approach Based on COLREGs Considering the Effects of the Wind and Wave"

_jmse, doi:10.3390/jmse11091831_

Round 1
Reviewer 1 Report
The aim of this article was to conduct a comprehensive study of unmanned vessel collision avoidance algorithm by dynamic window approach based on COLREGs considering the effects of the wind and wave. It was noted that only few existing DWA algorithms consider the COLREGs rule, including the CCDWA algorithm. However, the CCDWA algorithm has insufficient flexibility, and it is too rigid during emergency collision avoidance situations. In this study, the limitation of the sampling area was first changed to include prioritizing the sampling area to improve the flexibility of the CCDWA algorithm. Different priorities were given to the port and starboard speed sampling areas in the face of different collision avoidance scenarios. Subsequently, the original speed function was replaced by an improved speed formula that considers a windy and wavy environment, thereby allowing the USV to cleverly use the wind speed, wind direction, and waves to avoid collision and save time. The experimental results showed that the improved algorithm can consider windy and wavy environments and follow the COLREGs rules for more flexible collision avoidance. Summing up, the whole paper is clear, well-presented and fully understandable. Figures and Tables included in the paper are well prepared. There is also a direct relationship between the title and the paper. The entire text constitutes a well-written and very interesting contribution for MDPI. I have no substantive comments and I appreciate the study. I wish the authors good luck and further success in their research.
Author Response
Thank you for agreeing with the content of our study, your encouragement is the greatest motivation in our future research, and finally express our gratitude to you once again, wishing you good luck in your work and a happy life.
Reviewer 2 Report
1) Instead of explaining used abbreviations, symbols and markings throughout the paper text, it will be very helpful to any reader that all of the mentioned is arranged and explained in one place – in the Nomenclature. At the moment, the reader is required to constantly turn back through the paper to find a meaning of many abbreviations, symbols and markings. A Nomenclature will resolve that problem, and the paper readability will be notably improved.
2) Figure 7 – this figure represents characteristics of the UDWA algorithm, not the basic DWA (in the figure title stands DWA only, which makes a confusion). Moreover, this figure should be explained in much more detail. From the presented description, it is not clear what exact novelties and benefits brings this improvement in the basic DWA algorithm.
3) Equation 5 – the effects of wind and waves are considered by using this equation only. Wind and waves are too complex to be accurately and precisely considered by one equation only. Therefore, the question is – what are accuracy and precision of the presented equation? Is there any possibilities that the presented equation can be validated in the real exploitation conditions? The Authors should present and explain some details related to this equation, because it is very hard to believe that only one equation can properly consider the effects of wind and waves.
4) Table 2 - Wave heights – are this randomly selected data, data calculated according to some kind of numerical model or the data based on the real wave heights at the observed area? I believe that wind speeds cover the entire range of wind speeds on the selected geographic location – but the wave heights definition is completely unclear.
5) Figure 21 – (b) and (c) parts of this figure should not have the same title.
6) At the end of the Conclusions section should be presented guidelines for further research and parameters which can additionally be considered (and implemented) in the improved UDWA algorithm. Moreover, it can be recommended that the Authors add in the Conclusions section at least some of the most important results (its exact values) obtained in this research.
7) DWA algorithm improvement performed in this paper (to obtain UDWA) seems to be too simple and too general (at least in my opinion). The actual improvements were that the Authors assigned different priorities to two different velocity sampling regions for different encounter situations and that the effects of wind and waves are considered by using one general equation only. At the moment, I don’t see sufficient scientific contribution in the implementation of two above mentioned elements in the existing DWA algorithm. UDWA algorithm clearly shows better results than DWA, but there is no any confirmation or evidence that these results are sufficient. I hope that the Authors can extend their explanations and clearly evaluate novelty and paper contribution to the existing research field.
8) The English is understandable, but it can be improved in many sentences or whole paper parts. Please, perform a careful check of the English throughout the paper text.
9) The List of the References can be extended with more recent literature from this research field.
Final remarks: This is an interesting paper. However, the above mentioned elements should be resolved, added and properly explained during the revision process.
Minor editing of English language required.
Reviewer 3 Report
The article is about an improved DWA algorithm with COLREG rules and external environmental effects.
The article is well structured and very clearly presented. I have only one suggestion: add a "Discussion" section, because this is a crucial part where you interpret your simulation results, explain the context, and give insights into the implications of your study. Part of sections 4 and 5 can be moved to the "Discussion" section, and in the conclusion I suggest adding some thoughts on further research: based on the gaps and unanswered questions your study uncovered, suggest possible areas for future research. This shows that you are thinking beyond your current research and encourages further exploration of the topic.
Regards
Round 2
Reviewer 2 Report
The Authors have performed all mentioned and recommended additions/corrections/improvements.
Additional explanations were very helpful, especially in terms of the scientific novelty - I believe that novelty is now clear and obvious to anyone.
After proper revision, I have no more concerns related to this paper.
The paper should be published in a presented (revised) form.
My congratulations to the Authors.